# RotRNN: Modelling Long Sequences with Rotations

## Abstract

Linear recurrent neural networks, such as State Space Models (SSMs) and Linear Recurrent Units (LRUs), have recently shown state-of-the-art performance on long sequence modelling benchmarks. Despite their success, their empirical performance is not well understood and they come with a number of drawbacks, most notably their complex initialisation and normalisation schemes. In this work, we address some of these issues by proposing RotRNN – a linear recurrent model which utilises the convenient properties of rotation matrices. We show that RotRNN provides a simple and efficient model with a robust normalisation procedure, and a practical implementation that remains faithful to its theoretical derivation. RotRNN also achieves competitive performance to state-of-the-art linear recurrent models on several long sequence modelling datasets.

## 1 Introduction

Long sequence modelling is a notoriously difficult domain in machine learning due to the need to capture long-range dependencies between input data. Typical sequence models, such as Transformers (Vaswani et al., 2017) and Recurrent Neural Networks (RNNs) (Rumelhart et al., 1987; Hochreiter & Schmidhuber, 1997; Cho et al., 2014; Koutnik et al., 2014), fail to perform well on these tasks. In the case of Transformers this is due to poor inductive biases and quadratic scaling of computational complexity with sequence length, and in the case of non-linear RNNs it is caused by vanishing and exploding gradients. Recently, State Space Models (SSMs) (Gu et al., 2021; Smith et al., 2023; Gupta et al., 2022a) have emerged as the state-of-the-art framework for learning on long-sequences of data. The S4 model (Gu et al., 2021), inspired by linear time invariant dynamical systems, utilises a linear recurrent layer with HiPPO initialisation (Gu et al., 2020) to solve the vanishing and exploding gradient problem of classical RNNs. Moreover, the computational complexity of S4 scales linearly in time with sequence length, and thus circumvents the quadratic computational scaling of Transformers. Interestingly, the linear recurrence of S4 hidden state can be viewed as both a linear RNN for fast inference and as a Convolutional Neural Network (CNN) for efficient parallel training.

Despite the mathematical elegance of the S4 derivation, the question of whether such careful initialisation is required remains open. Indeed, several subsequent works suggested that the specific initialisation and discretisation methods that theoretically motivated S4 may not be necessary for highly performant SSMs (Gupta et al., 2022a;b; Gu et al., 2022; Smith et al., 2023; Orvieto et al., 2023). These findings led to the development of the Linear Recurrent Unit (LRU) (Orvieto et al., 2023), which showed that competitive empirical performance on long sequence modelling tasks can be achieved by making some small modifications to a standard linear RNN, without using the theoretical insights from SSMs. While the LRU is conceptually simpler than prior works, the theoretical motivation does not necessarily reflect the practical implementation of the algorithm, leading to more unanswered questions as to why SSMs and the LRU are stable and performative on long sequence modelling tasks.

The LRU and other linear recurrent layers have been used as basic building blocks for more complex sequence modelling architectures (Gu & Dao, 2023; De et al., 2024). Motivated by the wide adoption of such models (Zhu et al., 2024; Ma et al., 2024; Xing et al., 2024; Patro & Agneeswaran, 2024), we extend the body of work on linear RNNs by proposing a novel recurrent block, deriving a conceptually simple but mathematically principled way of performing linear recurrence by

using rotation matrices. By parameterising the recurrent state matrix as a rotation, we are able to provide more stable normalisation than prior works, and allow for a robust implementation that faithfully reflects its theoretical motivation. Importantly, our model remains consistent with the theory throughout training, and is not purely used to find a good initialisation of the recurrent matrix (Gu et al., 2021; Gupta et al., 2022a; Smith et al., 2023). Moreover, we derive a mathematical equivalence between a special case of the LRU and our model, in the hope that it will help shed light on some of the LRU's internal mechanics. We summarise our main contributions as follows:

- We propose a novel linear RNN layer where the recurrent state matrix $A$ **is parameterised as a rotation**, with the resulting algorithm faithfully reflecting the theoretical motivation throughout training.

- We present a method for computing **efficient matrix powers of parametric rotations** to enable fast linear recurrence with rotation matrices.

- We use this new formulation to derive a principled normalisation procedure which retains a **constant expected hidden state magnitude** that holds throughout training.

- We show **competitive performance with the state-of-the-art** on long sequence benchmarks, and show that the improved normalisation of our model holds on practical tasks.

## 2 BACKGROUND

### 2.1 STATE SPACE MODELS

SSMs (Gu et al., 2021; Smith et al., 2023; Gupta et al., 2022a) are derived from time-invariant continuous-time linear ordinary differential equations (ODEs), of the form

$$\dot{x}(t) = A'x(t) + B'u(t)$$
$$y(t) = Cx(t) + Du(t)$$

where $B' \in \mathbb{R}^{\mathcal{D}_x \times \mathcal{D}_u}$ is the input matrix, $A' \in \mathbb{R}^{\mathcal{D}_x \times \mathcal{D}_x}$ is the state matrix, $C \in \mathbb{R}^{\mathcal{D}_y \times \mathcal{D}_x}$ and $D \in \mathbb{R}^{\mathcal{D}_y \times \mathcal{D}_u}$ are the output matrices and $u(t) \in \mathbb{R}^{\mathcal{D}_u}$ is the continuous-time input.

Under a constant sampling rate with a given stepsize $\Delta > 0$, such systems can be discretised using Zero-Order Hold (ZOH) or Bilinear discritisation. Under the ZOH method, the resulting discrete system can be expressed by the following recursion:

$$x_t = Ax_{t-1} + Bu_t$$
$$y_t = Cx_t + Du_t \tag{1}$$

where $u_t$ are the sampled input signals and $A = \exp(\Delta A')$ and $B = (A - I)A'^{-1}B'$ are discretised versions of the state and input matrices, respectively. Importantly, this recurrence relation can also be unrolled and written as a convolution over the inputs,

$$x_t = \sum_{k=1}^{t} A^{t-k}Bu_k. \tag{2}$$

This duality of recurrence and convolution allows for efficient parallel computation of sequence outputs during training, and fast state updating during inference. Equation 1 is the foundation of the SSM layer in S4 (Gu et al., 2021) and its variants (Gupta et al., 2022a; Smith et al., 2023; Gu & Dao, 2023). The matrix $A$ is initialised using HiPPO theory (Gu et al., 2020), whose derivation follows from the theory of optimal polynomial projections.

### 2.2 LINEAR RECURRENT UNITS

Instead of discretising a continuous-time ODE, the LRU (Orvieto et al., 2023) achieves competitive empirical performance with clever parameterisation and normalisation of linear RNNs. The derivation of the LRU is motivated by the observation that the recurrent matrix $A \in \mathbb{R}^{\mathcal{D}_x \times \mathcal{D}_x}$ can be written (up to arbitrarily small perturbation of the entries (Axler, 2024)) as

$$A = P\Lambda P^{-1} \tag{3}$$

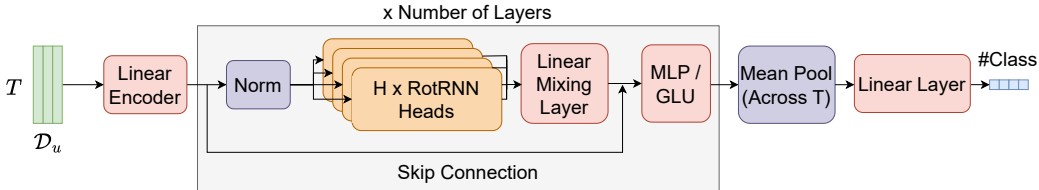

Figure 1: Full neural network architecture of the RotRNN. Here $T$ denotes the length of the input sequence, and $\mathcal{D}_u$ denotes the number of channels in the input data.

where $\Lambda = \mathrm{diag}(\lambda_1, \ldots, \lambda_{\mathcal{D}_x}) \in \mathbb{C}^{\mathcal{D}_x \times \mathcal{D}_x}$ is the diagonal matrix of eigenvalues and $P \in \mathbb{C}^{\mathcal{D}_x \times \mathcal{D}_x}$ is a complex-valued invertible matrix of eigenvectors. This diagonalised parameterisation is necessary to allow for fast computation of matrix powers, which is required in linear recurrent models (see Equation 2). Premultiplying both sides of Equation 2 by $P^{-1}$, and plugging in Equation 3, we obtain

$$\tilde{x}_t = \sum_{k=1}^{t} \Lambda^{t-k} \tilde{B} u_k$$
$$y_t = \tilde{C} \tilde{x}_t + D u_t \tag{4}$$

where $\tilde{x}_t = P^{-1} x_t$, $\tilde{B} = P^{-1} B$, $\tilde{C} = CP$. The LRU aims to directly learn the matrices $\tilde{B}$ and $\tilde{C}$, along with the eigenvalues $\lambda_j = \nu_j e^{i\theta_j}$, for learnable parameters $\nu_j, \theta_j \in \mathbb{R}$.

## 2.3 DRAWBACKS OF PRIOR WORKS

These prior methods for tackling long sequence modelling with linear recurrence have several drawbacks. In the case of SSMs, a complicated theoretical derivation based on polynomial projections is required to initialise the recurrent HiPPO matrix. However, this is purely used as an initialisation procedure. The inner workings of SSM models throughout training is not well understood, and more research is needed to uncover why deviation from optimal polynomial projections of functions improves results and remains stable throughout learning. Indeed, as shown in the LRU, it is in fact unnecessary to use such theoretical motivation to achieve strong performance on long sequence benchmarks.

What actually goes on under the hood of the LRU, though, is also not entirely clear. The motivation of the LRU stems from constraining classical linear RNNs, but this is not fully reflected in the final proposed algorithm. Firstly, eigenvalues of the real matrix $A$ come in conjugate pairs, but this is not enforced in the LRU (as it is in, for example, S5 (Smith et al., 2023)). Moreover, there is no constraint ensuring that $y_t = \tilde{C} \tilde{x}_t = CPP^{-1} x_t = C x_t$ is real-valued – i.e. that the $P$ and $P^{-1}$ components of the learned $\tilde{C}$ and $\tilde{x}_t$ are consistent. Instead, the authors simply take the real part of the resulting complex $y_t$. Additionally, the LRU is normalised (at initialisation) by ensuring expected convergence in the limit of infinite sequence length (see 4.1), but in practice a learnt normaliser is used, and does not necessarily result in desirable or consistent hidden state magnitudes (Figure 3).

In this work, we aim to overcome this mis-match between theoretical motivation and practical implementation by proposing a novel linear recurrent model using rotation matrices. Our algorithm is conceptually simple to understand, efficient to compute, robust to exploding hidden state norms, and, importantly, faithfully reflects the mathematical principles that underpin its motivation.

## 3 RotRNN

The key component of our model, which we call the **Rot**ational **R**ecurrent **N**eural **N**etwork (RotRNN), is the parameterisation of the recurrent state matrix $A$ as a rotation matrix. The reason for this choice is three-fold:

i) Rotation matrices can be generated smoothly from any real-valued matrix, making them robust to initialisation and easy to constrain during training (Section 3.1).

ii) Rotations can be easily decomposed for fast computation of matrix powers, which is needed in linear RNNs (Section 3.2).

iii) The orthogonality of rotation matrices, and the fact that their eigenvalues lie on the unit circle, allows us to derive a simple normalisation scheme that retains a constant expected hidden state norm throughout training (Section 3.3).

## 3.1 Parameterising Rotation Matrices

We first consider the problem of parameterising $A$ as a rotation matrix in the linear RNN formulation. The group comprising all rotation matrices in $\mathbb{R}^{N \times N}$ is known as the special orthogonal group, $SO(N)$, defined as follows:

$$SO(N) = \{Q \mid Q \in \mathbb{R}^{N \times N}, Q^\top Q = QQ^\top = I, \det(Q) = 1\}. \tag{5}$$

To ensure that a learnable matrix $A$ remains in $SO(N)$ throughout training, instead of learning a constrained $A$ directly we can learn a general weight matrix $M \in \mathbb{R}^{N \times N}$ and smoothly map $M$ onto $SO(N)$. To do this, we make use of the following lemma:

**Lemma 1.** *Let $M \in \mathbb{R}^{N \times N}$, let $S = M - M^\top$, and define $\exp(S) := \sum_{k=0}^\infty \frac{1}{k!} S^k$ as the matrix exponential. Then $A = \exp(S) \in SO(N)$.*

*Proof.* See App. A.1. □

The matrix exponential map is surjective from skew-symmetric matrices onto $SO(N)$ (Rohan, 2013), so this parameterisation is sufficiently general for learning arbitrary rotation matrices.

## 3.2 Efficient Rotational Recurrence

Unfortunately, computing the convolutional form of linear recurrent layers (Equation 2) during training involves taking matrix powers of $A$, which is generally slow for high-dimensional, dense matrices. To make this computation more efficient, we utilise the structure of rotation matrices, using the following result to decompose the dense rotation matrix into block-diagonal form.

**Lemma 2.** *Let $P \in O(N)$ be an orthogonal matrix and let $\Theta \in SO(N)$ be a block-diagonal rotation matrix. Then $A = P\Theta P^\top \in SO(N)$. Moreover, any rotation matrix can be written in this form (Gallier & Xu, 2003).*

*Proof.* See App. A.2. □

The matrix $\Theta$ has $N/2$ blocks of the form $\begin{pmatrix} \cos\theta_i & -\sin\theta_i \\ \sin\theta_i & \cos\theta_i \end{pmatrix}$ along the diagonal if $N$ is even, and $\frac{N-1}{2}$ blocks if $N$ is odd with the remaining value on the diagonal being 1, where $\theta_i \in [0, 2\pi]$, $i = 1, \ldots, \lfloor \frac{N}{2} \rfloor$, are axis-aligned rotation angles in the space projected onto by $P$. When computing matrix powers, the dense orthogonal matrices $P$ and $P^\top$ cancel out leaving $A^k = P\Theta^k P^\top$, where the blocks of $\Theta^k$ are of the form $\begin{pmatrix} \cos k\theta_i & -\sin k\theta_i \\ \sin k\theta_i & \cos k\theta_i \end{pmatrix}$. Therefore, by learning an orthogonal matrix $P$ and set of rotation angles $\boldsymbol{\theta} = \{\theta_1, \ldots, \theta_{\lfloor \frac{N}{2} \rfloor}\}$ directly, we can easily generate rotation matrices that are amenable to computing fast matrix powers.

It is important to note that, unlike $SO(N)$, there is no smooth surjective map from real square matrices onto $O(N)$, the General Orthogonal group:

$$O(N) = \{Q \mid Q \in \mathbb{R}^{N \times N}, Q^\top Q = QQ^\top = I, \det(Q) = \pm 1\}.$$

It is therefore hard in practice to parameterise $P$ such that the space of learnable $P$ covers the entire group of orthogonal matrices. Instead, we learn a general weight matrix $M \in \mathbb{R}^{N \times N}$, and use Lemma 1 to obtain $P = \exp(M - M^\top) \in SO(N) \subset O(N)$. While this means that $A = P\Theta P^\top$ may not be surjective onto $SO(N)$, we find it is sufficiently general to achieve good results in practice (see Section 5).

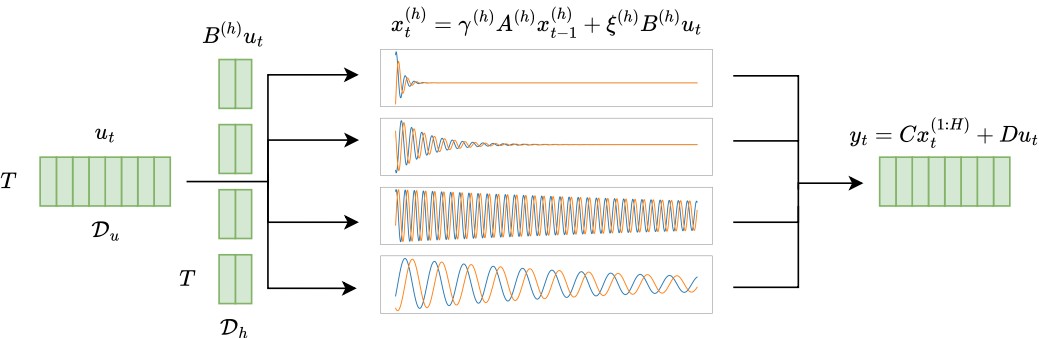

Figure 2: A visualisation of the mulit-headed RotRNN layer outlined in Section 3.4. The $\mathcal{D}_u$-dimensional input sequence $u_t$, $t = 1, \ldots, T$, is projected onto each of the $H$ heads of dimension $\mathcal{D}_h$ by the $B^{(h)}$ matrices. Each head then independently performs a linear recurrence with different rotations and decay scales. The outputs of each head are concatenated and mixed linearly to form the final $\mathcal{D}_u$-dimensional output $y_t$.

### 3.3 NORMALISATION

Now that we have a way to parameterise our recurrent state matrix for efficient recurrence, we must turn to the problem of normalising the recurrent state. This is a key ingredient of long-range recurrent networks, as it ensures the hidden state does not vanish or explode across long sequences. In prior works, this is either done implicitly by discretisation, as is the case in S4 (Gu et al., 2021), or explicitly with a normalisation constant, as in the LRU (Orvieto et al., 2023). In this work, we take an explicit approach to normalisation, leveraging the properties of the rotation matrix $A$ to derive a normalisation constant that retains a constant expected norm of the recurrent state at all times throughout training.

We define the hidden state recurrence of the RotRNN as

$$x_t = \alpha(\gamma A x_{t-1} + B u_t) \tag{6}$$

where $\alpha \in \mathbb{R}$ is a normalisation constant and $\gamma \in (0, 1)$ is a learnable scalar decay factor which controls the trade-off in importance between recent and distant-past input values.

**Lemma 3.** *Following Orvieto et al. (2023), let the inputs $u_t$ be sampled i.i.d., with mean 0 and variance I. Then for any constant c, if $\mathbb{E}\left[||x_1||^2\right] = c$ and $\alpha = \frac{1}{\sqrt{c\gamma^2 + \mathrm{Tr}[B^\top B]}}$ we have that $\mathbb{E}\left[||x_t||^2\right] = c$ for all timesteps t.*

*Proof.* We will prove this by induction. Under the assumption $\mathbb{E}\left[||x_1||^2\right] = c$, we only need to prove that $\mathbb{E}\left[||x_{t-1}||^2\right] = c \implies \mathbb{E}\left[||x_t||^2\right] = c$ if $\alpha$ is as stated above. Taking the expected square norm of Equation 6, we have

$$\mathbb{E}\left[||x_t||^2\right] = \alpha^2 \left(\gamma^2 \mathbb{E}\left[||x_{t-1}||^2\right] + \mathbb{E}\left[u_t^\top B^\top B u_t\right] + 2\gamma \mathbb{E}\left[x_{t-1}^\top A^\top B u_t\right]\right) \tag{7}$$

$$= \alpha^2 \left(\gamma^2 c + \mathrm{Tr}\left[B^\top B \,\mathbb{E}\left[u_t u_t^\top\right]\right]\right) \tag{8}$$

$$= \alpha^2 \left(\gamma^2 c + \mathrm{Tr}\left[B^\top B\right]\right) \tag{9}$$

Where in the first line we used the orthogonal property of the rotation matrix $AA^\top = A^\top A = I$, and in the second line we used the induction assumption that $\mathbb{E}\left[||x_{t-1}||^2\right] = c$ and that for i.i.d inputs $x_{t-1}$ and $u_t$ are uncorrelated. Finally, setting $\alpha = \frac{1}{\sqrt{c\gamma^2 + \mathrm{Tr}[B^\top B]}}$ gives $\mathbb{E}\left[||x_t||^2\right] = c$ as required. $\qquad\square$

In practice, however, we find that this naïve method of normalisation is not entirely satisfactory. Unrolling Equation 6 into its convolutional form we obtain

$$x_t = \sum_{k=1}^{t} \alpha^{t+1-k} \gamma^{t-k} A^{t-k} B u_k \tag{10}$$

where we can see that the normalisation constant $\alpha$ is raised to the power $t + 1 - k$. When $\alpha < 1$ we therefore observe that the weighting of early inputs in the sequence goes to zero exponentially, and the model quickly forgets all but the very recent past. Since we desire that the recurrent decay is controlled only by $\gamma$, we instead enforce $\alpha = 1$, shifting all normalisation into the matrix $B$. This requires that $\mathrm{Tr}\left[B^\top B\right] = 1 - c\gamma^2$, which can be achieved by simply re-scaling $B$ with the coefficient

$$\xi := \sqrt{\frac{1 - c\gamma^2}{\mathrm{Tr}\left[B^\top B\right]}}. \tag{11}$$

Our final expression for the recurrent and convolutional forms of the RotRNN hidden state is thus

$$x_t = \gamma A x_{t-1} + \xi B u_t \quad \Leftrightarrow \quad x_t = \xi \sum_{k=1}^{t} \gamma^{t-k} A^{t-k} B u_k \tag{12}$$

which avoids the problem of unwanted exponential decay. In practice we find that setting $c = 1$ is sufficient to achieve stable and robust normalisation, even in deep, multi-layer networks (Figure 3).

### 3.4 Multi-Head Decay

The price of having a constant expected hidden state norm in our derivation is that the decay factor $\gamma$ must be scalar. We find, however, that this does not generalise well to problems that require retaining information from horizons at different scales. We address this by running $H$ independent low-dimensional RotRNN heads in parallel (see Figure 2), each with a unique set of $A^{(h)} \in \mathbb{R}^{\mathcal{D}_h \times \mathcal{D}_h}, B^{(h)} \in \mathbb{R}^{\mathcal{D}_h \times \mathcal{D}_u}, \gamma^{(h)} \in (0, 1)$ parameters, where we set $\mathcal{D}_h = \frac{\mathcal{D}_x}{H}$. The output projection,

$$y_t = C x_t^{(1:H)} + D u_t, \tag{13}$$

where $C \in \mathbb{R}^{\mathcal{D}_u \times \mathcal{D}_x}, D \in \mathbb{R}^{\mathcal{D}_u \times \mathcal{D}_u}$, and $x_t^{(1:H)}$ is the concatenation of the $t^{\text{th}}$ hidden state from each head, can be viewed as a linear mixing layer. This enables the model to share information from the different rotation phase and decay horizon recurrences from each head, which is critical in tasks which require learning both long and short range dependencies between inputs. We note that, although we use $\mathcal{D}_h = \frac{\mathcal{D}_x}{H}$ in our experiments and for ease of mathematic notation, this set-up is easily scalable to any desired head dimension $\mathcal{D}_h \geq 2$.

## 4 Analysis of RotRNN and Prior Work

### 4.1 Linear Recurrent Units

The RotRNN algorithm proposed in this paper is inspired in part by the LRU (Orvieto et al., 2023). As well as sharing superficial similarities in the structure of the recurrent layer, more formal comparisons can be drawn between the two architectures. In this section, we derive a mathematical equivalence between a special case of the LRU and the RotRNN, and compare the normalisation procedures of the two models.

**Multihead RotRNN as a special case of the LRU** In practice, ignoring skip connections, the LRU recurrent layer has 3 parameter matrices: $\Lambda$, a diagonal matrix of complex eigenvalues; $\tilde{B}$, a dense, complex input matrix; $\tilde{C}$, a dense, complex linear output projection. Consider the case where the learned eigenvalues of $A$ come in complex-conjugate pairs. In this case, the matrix $\Lambda$ can be written as a block diagonal matrix of $\begin{pmatrix} \nu_j \cos\theta_j & -\nu_j \sin\theta_j \\ \nu_j \sin\theta_j & \nu_j \cos\theta_j \end{pmatrix}$, with $\tilde{B}$ and $\tilde{C}$ real matrices (see App. E of Orvieto et al. (2023) for details). If one assumes that $\tilde{B} = P^\top B$ and $\tilde{C} = CP$ for some block-diagonal orthogonal matrix $P$, then, up to normalisation, this is algebraically equivalent to the multi-head RotRNN with $H = \frac{\mathcal{D}_x}{2}$ heads and the dimension of each head is $\mathcal{D}_h = 2$. This is because the parallel headed structure of the RotRNN can be viewed as one big block-diagonal recurrent layer, with corresponding dimensions of $B$ that project onto each head being normalised independently, and the corresponding decay factors $\gamma^{(h)}$ modulating the eigenvalue magnitude as does $\nu_j$ in the LRU.

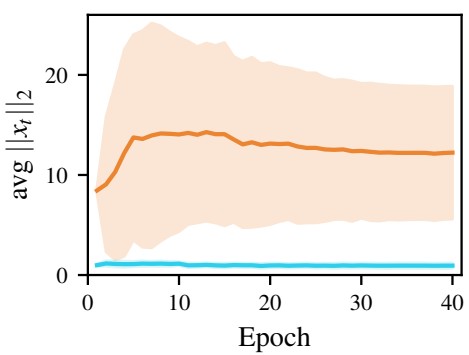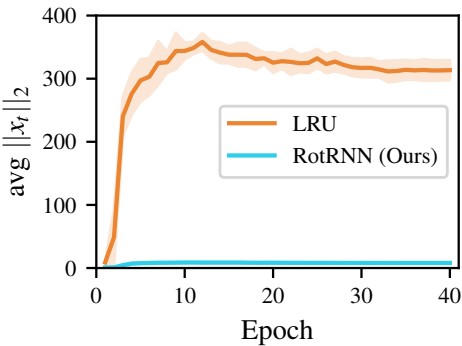

(a) Mean $||x_t||_2$ across different recurrent layers during training in an 8-layer model.

(b) Mean $||x_t||_2$ in a single layer network across 5 random seeds of training.

Figure 3: Average hidden state norm across training on ListOps for the LRU (Orvieto et al., 2023) and RotRNN. The standard deviation of the means is plotted in the error bars. We note that the error bars for RotRNN are present, but are mostly too small to be visible.

Note that, despite being algebraically possible in the LRU theoretical framework, this exact special case of the LRU is unlikely to occur naturally in the practical algorithm proposed in Orvieto et al. (2023) (outlined in Section 2.2). This is because the model would be required to learn that $\lambda_{1:\mathcal{D}_x}$ come in conjugate pairs, and the learned $P$ components of the input and output matrices to be the diagonalising matrix of the block-diagonal $A^{(1:H)} \in \mathbb{R}^{\mathcal{D}_x \times \mathcal{D}_x}$ of the RotRNN. However, we still believe that this view of the LRU as a decayed rotation-based block can help to uncover the mechanics of its linear recurrence.

**Normalisation differences** To compare the normalisation methods between the two algorithms, we give a brief overview of the derivation for the LRU normalisation constant found in Orvieto et al. (2023). Assuming white-noise input, one can calculate the expected norm of the LRU hidden state:

$$\mathbb{E}\left[||x_t||^2\right] = \mathbb{E}\left[\left(\sum_{i=1}^{t} \Lambda^i \tilde{B} u_{t-i}\right)^* \left(\sum_{j=1}^{t} \Lambda^j \tilde{B} u_{t-j}\right)\right]$$

$$= \sum_{i=1}^{t} \sum_{j=1}^{t} \text{Tr}\left[\mathbb{E}\left[u_{t-i}^* \tilde{B}^* {\Lambda^i}^* \Lambda^j \tilde{B} u_{t-j}\right]\right]$$

$$= \sum_{i=1}^{t} \sum_{j=1}^{t} \text{Tr}[\tilde{B}\mathbb{E}\left[u_{t-j} u_{t-i}^*\right] \tilde{B}^* {\Lambda^i}^* \Lambda^j] = \sum_{i=1}^{t} \text{Tr}[\tilde{B}\tilde{B}^* {\Lambda^i}^* \Lambda^i] \quad (14)$$

Since $\Lambda$ is diagonal, Equation 14 can be re-written into a summation of terms $b_k |\lambda_k|^2$, where $b_k$ is the squared norm of each row of $\tilde{B}$, and $\lambda_k \in \mathbb{C}$ is the $k$'th diagonal entry of $\Lambda$. Hence the expected norm becomes:

$$\mathbb{E}\left[||x_t||^2\right] = \sum_{i=1}^{t} \sum_{k=1}^{N} b_k |\lambda_k|^{2i} = \sum_{k=1}^{N} b_k \sum_{i=1}^{t} |\lambda_k|^{2i} \xrightarrow{t=\infty} \sum_{k=1}^{N} b_k \frac{1}{1 - |\lambda_k|^2}$$

To ensure a finite expected norm in the limit of $t \to \infty$, one can normalise the rows of $\tilde{B}$ element-wise by $\sqrt{1 - |\lambda_k|^2}$. This is similar to the RotRNN normalisation outlined in Section 3.3, as the decay parameters $\gamma^{(h)}$ in each head essentially control the eigenvalue magnitude of the $h^{\text{th}}$ recurrent state matrix. Conversely, in RotRNN the added $\text{Tr}\left[B^\top B\right]$ term applied independently across heads ensures that the expected norm of the recurrent state remains constant throughout the entire sequence, providing stronger guarantees than expected convergence at infinite sequence length. We do, however, use the same white-noise input assumption as in (Orvieto et al., 2023), which we hope to be able to overcome in future work for normalisation guarantees on more general input types.

Table 1: Test accuracy on the LRA benchmark tasks. We follow the standard training procedures from Gu et al. (2021). Unless otherwise specified, we report the results of baseline methods from their respective citations.

| Model (Input Length) | ListOps (2,048) | Text (4,096) | Retrieval (4,000) | Image (1,024) | Pathfinder (1,024) | Path-X (16,384) | Avg. |
|---|---|---|---|---|---|---|---|
| S4 (Gu et al., 2021) | 59.6 | 86.8 | 90.9 | **91.1** | 94.2 | 96.4 | 86.5 |
| S4D (Gu et al., 2022) | 60.5 | 86.2 | 89.5 | 88.2 | 93.1 | 92.0 | 84.9 |
| Liquid-S4 (Hasani et al., 2023) | **62.8** | 89.0 | 91.2 | 89.5 | 94.8 | 96.7 | 87.3 |
| S5 (Smith et al., 2023) | 62.2 | 89.3 | **91.4** | 90.1 | **95.3** | **98.6** | **87.8** |
| LRU (Axler, 2024) | 60.2 | 89.4 | 89.9 | 89.0 | 95.1 | 94.2 | 86.3 |
| LRU (Our Reprod.) | 57.9 | 89.4 | 89.4 | 85.2 | 90.0 | 92.8 | 84.1 |
| RotRNN (Ours) | 61.1 | **89.6** | 89.9 | 85.9 | 93.0 | 89.2 | 84.8 |

## 4.2 State Space Models

The relationship between RotRNN and other SSMs, namely S4 (Gu et al., 2021) and S5 (Smith et al., 2023), is perhaps less formally equivalent, but we can still draw comparisons between the structures of the recurrent layers. The S4 recurrence can be viewed as a stack of single-input-single-output (SISO) SSMs, whereby independent recurrent layers operate on each channel of the vector-valued input. The outputs of these independent SSMs are concatenated and passed through a "mixing layer" to combine information. S5, on the other hand, uses a single multi-input-multi-output (MIMO) SSM as its recurrent layer, and as such does not require a separate mixing layer to share information across dimensions. The input-output structure of RotRNN sits somewhere in-between S4 and S5. The use of multiple independent RotRNN heads outlined in Section 3 can be viewed as a stack of independent MIMO recurrent layers, in which the user may specify both the number of heads and the dimension of each head separately. Multiplication with the output matrix $C$ can be viewed as a linear mixing layer, combining information from the hidden states of each independent head. For more details on the SISO and MIMO views of S4 and S5, we direct the reader to Smith et al. (2023).

## 5 Experiments

We evaluate the RotRNN on several long sequence modelling datasets, comparing performance to state-of-the-art linear recurrent sequence models. RotRNN performs competitively throughout, but particularly excels on very discrete input data (such as text). We also empirically analyse our normalisation procedure compared to that of the LRU across both deep and single-layer networks.

### 5.1 Long Range Arena

We evaluate the performance of RotRNN on Long Range Arena (LRA) (Tay et al., 2021), a set of 6 sequence modelling tasks with sequence lengths between 1K and 16K tokens and varying data modalities. Tab. 1 shows the results for RotRNN and other linear recurrent models for comparison, reporting the test accuracy at the highest validation accuracy throughout training. Overall, we find that RotRNN performs competitively with other state-of-the-art linear recurrent models. In particular, we find that our model performs best on domains with more discrete input data, such as ListOps, Text and Retrieval, achieving the highest score of all the baselines in the IMDB classification task (Text). However, we also note that RotRNN falls short of some of the baselines on the pixel-level image tasks, such as Path-X and Cifar.

**Hidden State Norms** We plot the mean hidden state norm of the recurrent layers of the LRU and RotRNN throughout training on the ListOps dataset in Figure 3. We find that the hidden states of the RotRNN have an almost constant magnitude throughout training, with very little variance across layer depth or random initialisations. To contrast this, the LRU hidden state norms vary wildly across layer depth, and take far longer to converge (if ever) to a reasonably constant magnitude throughout

training. Moreover, we see that the size and variance of the norms are significantly larger that those of the RotRNN across random seeds in a single-layer network.

## 5.2 RAW SPEECH CLASSIFICATION

Since LRA (Tay et al., 2021) is partly a synthetic benchmark, we further evaluate RotRNN on a more natural long-sequence classification task: the Speech Commands dataset (Warden, 2018). This dataset contains 1s waveforms of 35 spoken English words, sampled at 16kHz. The task is to classify the word from its given sampled waveform. The results are displayed in Tab. 2. We find that RotRNN performs identically well to the LRU (Orvieto et al., 2023), with a similar number of parameters. It also remains very competitive with other, more theoretically complex deep state space models.

Table 2: Test accuracy on the 35-way Speech Commands classification task (Warden, 2018). Unless otherwise specified, we report the results of baseline methods from their respective citations.

| Model (Input Length) | Params. | 16kHz (16,000) |
|---|---|---|
| S4 (Gu et al., 2021) | 307K | 96.1 |
| S4D (Gu et al., 2022) | 306K | 95.8 |
| Liquid-S4 (Hasani et al., 2023) | 224K | **96.8** |
| S5 (Smith et al., 2023) | 280K | **96.8** |
| LRU (Our Reprod.) | 283K | 95.2 |
| RotRNN (Ours) | 284K | 95.2 |

## 6 RELATED WORK

**Orthogonal Recurrent Networks** The use of orthogonal and rotation matrices in recurrent networks has been explored previously in non-linear RNNs. Unitary RNNs (uRNNs) (Arjovsky et al., 2016; Jing et al., 2017) prevent blow-up in long sequences by parameterising recurrent matricies with unitary matrices (an extension of orthgonality to the complex field) to ensure an eigenvalue magnitude of 1. This idea has since been applied without the need for complex numbers (Helfrich et al., 2018), and has inspired works replacing recurrent operators in LSTMs with rotations (Dangovski et al., 2019; Velici & Prügel-Bennett, 2021). These methods, however, still suffer from the drawbacks of classical non-linear RNNs, such as inefficient computation of recurrent states during training when compared to associative-scan based linear models.

**Building on Linear Recurrent Models** Linear recurrent layers have recently been used as building blocks to construct more complex long sequence models. Two representative examples include Griffin (De et al., 2024) built on top of the LRU, and Mamba (Gu & Dao, 2023) built on top of S4. The key to the success of both these models is *gating* – the ability to construct recurrent state matrices based on the current inputs to selectively control information flow. We believe that the RotRNN could be used as a drop-in replacement for the LRU in Griffin, or be used to perform alternative gating strategies with input dependent rotations, but we leave this direction for future work.

## 7 CONCLUSIONS AND FUTURE WORK

In this paper we propose RotRNN, a linear recurrent model that utilises the convenient properties of rotation matrices. We show that RotRNN performs competitively with the state-of-the-art on long-range sequence modelling benchmarks, while providing a conceptually simple and efficient algorithm. RotRNN remains faithful to its theoretical derivation throughout training, with a robust normalisation procedure that does not rely on complex initialisation.

In addition, we hope that the concrete comparisons between our model and the LRU drawn in Section 4 can shed new light on the inner workings of the LRU and similar algorithms as multi-headed rotation-based linear recurrent networks. We point future investigations towards integrating

RotRNN into more complex architectures to test its downstream capacity on other domains, and implementing input dependent rotation transitions for gated rotational recurrence.

## 8 REPRODUCIBILITY STATEMENT

We provide a number of elements in this paper to help improve the reproducibility of our results. Firstly, we describe in detail the methods used to construct the RotRNN in Section 3, and show a full neural network architecture used for our experiments in Figure 1. Moreover, we provide the hyperparameters used and details of the initialisation and parameterisation of learnable parameters in App. B. Finally, we provide a simplified JAX implementation of the RotRNN in App. C.

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

## A   PROOFS

In this section we provide proofs of all theorems and lemmas not proven in the main text.

### A.1   PROOF OF LEMMA 1

To prove Lemma 1, we must first state the formal definition of $SO(N)$.

**Definiton 4** (Special Orthogonal Group). *The Special Orthogonal group, $SO(N)$, is defined as*

$$SO(N) = \left\{ A \in \left( \mathbb{R}^{N \times N}, * \right) \mid A^\top A = AA^\top = I, \det(A) = 1 \right\}$$

*where the group operation $*$ denotes matrix multiplication.*

Hence, to prove Lemma 1, we must simply show that any matrix $M \in \mathbb{R}^{N \times N}$ under the respective transformation is both orthogonal and has determinant 1.

**Lemma 1.** *Let $M \in \mathbb{R}^{N \times N}$, let $S = M - M^\top$, and define $\exp(S) := \sum_{k=0}^\infty \frac{1}{k!} S^k$ as the matrix exponential. Then $A = \exp(S) \in SO(N)$.*

*Proof.* To prove orthogonality of $A$, we will use the well-known fact that for two square matrices $P, Q \in \mathbb{R}^{N \times N}$, if $PQ = QP$ then $\exp(P) \exp(Q) = \exp(P + Q)$. Since $S$ is skew-symmetric, we have that $S^\top = -S$ and hence $SS^\top = S^\top S = -S^2$. Moreover, since the matrix exponential is defined by a power series, we have that $\exp(S)^\top = \exp(S^\top)$. Putting these two together we get

$$AA^\top = \exp(S) \exp(S)^\top = \exp(S) \exp(S^\top) = \exp(S + S^\top) = \exp(S - S) = I \qquad (15)$$

and clearly the same is true for $A^\top A$. Hence, $A$ is orthogonal.

To prove that determinant is 1, we use Jacobi's formula, which states that for any square matrix $S$, $\det(\exp(S)) = \exp(\mathrm{Tr}\,[S])$. Since $S$ is skew-symmetric, we have that $\mathrm{Tr}\,[S] = 0$, and hence

$$\det(A) = \det(\exp(S)) = \exp(\mathrm{Tr}\,[S]) = \exp(0) = 1 \qquad (16)$$

$\square$

### A.2   PROOF OF LEMMA 2

**Lemma 2.** *Let $P \in O(N)$ be an orthogonal matrix and let $\Theta \in SO(N)$ be a block-diagonal rotation matrix. Then $A = P\Theta P^\top \in SO(N)$. Moreover, any rotation matrix can be written in this form (Gallier & Xu, 2003).*

*Proof.* We provide only a proof for the first statement, as it is the only one functionally relevant for the RotRNN to be valid, but the converse statement follows trivially from the Jordan form of orthogonal matrices. For the first statement, we must again show that $A$ is orthogonal with determinant 1. The first condition is satisfied due to the orthogonality of $P$ and $\Theta$ as follows

$$AA^\top = P\Theta P^\top P\Theta^\top P^\top = P\Theta\Theta^\top P^\top = PP^\top = I \qquad (17)$$

and similarly for $A^\top A$.

The second condition is satisfied by noting that all matrices in $SO(N)$ have determinant 1, all matrices in $O(N)$ have determinant $\in \{\pm 1\}$, and $\det(P) = \frac{1}{\det(P^\top)}$ for orthogonal $P$. Hence,

$$\det(A) = \det(P\Theta P^\top) = \det(P) \det(\Theta) \det(P^\top) = 1 \qquad (18)$$

$\square$

## B   IMPLEMENTATION DETAILS

We implement the RotRNN in JAX due to its associative scan operator for fast, parallel computation of the recurrent states across long sequence. Figure 1 provides an overview of the entire RotRNN architecture. In the following subsections, we will discuss implementation details that make our code efficient and provide the hyperparameters used in our experiments. A simplified JAX implementation of RotRNN is given in App. C.

### B.1 ROTATION MATRIX MULTIPLICATION

We use the associative scan operation to implement the RotRNN layer with an efficient matrix multiplication algorithm, making use of the special structure of our rotation matrix. In particular, since our rotation matrix factorises into $A = P\Theta P^\top$, the orthogonality of $P$ means we only need to multiply by the block-diagonal matrix $\Theta$ in the associative scan. Multiplying a vector by a block diagonal matrix can be implemented efficiently, $\Theta x$ can be equivalently computed as

$$\Theta x = \begin{bmatrix} x_1 \\ x_2 \\ x_3 \\ x_4 \\ \vdots \\ x_{\mathcal{D}_x-1} \\ x_{\mathcal{D}_x} \end{bmatrix} \odot \begin{bmatrix} \cos\theta_1 \\ \cos\theta_1 \\ \cos\theta_2 \\ \cos\theta_2 \\ \vdots \\ \cos\theta_{\mathcal{D}_x//2} \\ \cos\theta_{\mathcal{D}_x//2} \end{bmatrix} + \begin{bmatrix} -x_2 \\ x_1 \\ -x_4 \\ x_3 \\ \vdots \\ -x_{\mathcal{D}_x} \\ x_{\mathcal{D}_x-1} \end{bmatrix} \odot \begin{bmatrix} \sin\theta_1 \\ \sin\theta_1 \\ \sin\theta_2 \\ \sin\theta_2 \\ \vdots \\ \sin\theta_{\mathcal{D}_x//2} \\ \sin\theta_{\mathcal{D}_x//2} \end{bmatrix} \qquad (19)$$

where $\odot$ denotes element-wise multiplication.

### B.2 HYPERPARAMETERS

In our experiments we use bidirectional RotRNN layers for Pathfinder and Path-X datasets, while for the rest of the datasets we use unidirectional layers. In the bidirectional layer we reuse the parameters $P$, $\Theta$ and $B$, while the matrix $C$ is different for the forward and the backward pass. We use batch normalisation for all of our experiments, and the number of layers $L = 6$ and number of heads $H = 32$ were the same for all experiments. The rest of the hyperparameters are described in Tab. 3. We select initial hyperparameters from the literature surrounding Linear Recurrent Networks and State Space Models, and performed small hyperparameter sweeps for ListOps, Text, Retrieval, Image and Speech Commands, grid-searching the hyperparameter spaces for $\gamma_{\text{init}}$, $\theta_{\text{init}}$ and learning rate. We tuned the hyperparameters for Pathfinder and Path-X manually. The total number of parameters in the resulting model is very similar to the baseline models (Gu et al., 2021; Smith et al., 2023; Orvieto et al., 2023; Gupta et al., 2022a; Hasani et al., 2023).

Table 3: Hyperprameters used for training on LRA and Speech Commands. $D$=model dimension, $N$=recurrent layer dimension, $GLR$=global learning rate, $LR$=recurrent layer learning rate, $B$=batch size, $WD$=weight decay, $\gamma_{\text{init}}$=initialisation range for $\gamma$, $\theta_{\text{init}}$=initialisation range for $\theta$.

| Dataset | $D$ | $N$ | $GLR$ | $LR$ | $B$ | $WD$ | Drop. | Iters. | $\gamma_{\text{init}}$ | $\theta_{\text{init}}$ |
|---|---|---|---|---|---|---|---|---|---|---|
| ListOps | 128 | 256 | 1e-3 | 1e-3 | 32 | 0.05 | 0.0 | 80K | [0.5, 0.999] | $[0, \pi/100]$ |
| Text | 256 | 192 | 1e-3 | 1e-3 | 32 | 0.05 | 0.1 | 50K | [0.5, 0.8] | $[0, \pi/10]$ |
| Retrieval | 128 | 256 | 1e-4 | 1e-5 | 32 | 0.01 | 0.1 | 50K | [0.5, 0.999] | $[0, 2\pi]$ |
| Image | 512 | 384 | 4.5e-3 | 1e-3 | 50 | 0.05 | 0.1 | 250K | [0.99, 0.999] | $[0, 2\pi]$ |
| Pathfinder | 192 | 256 | 4.5e-3 | 1e-3 | 64 | 0.03 | 0.05 | 500K | [0.1, 0.9999] | $[0, \pi/10]$ |
| PathX | 192 | 256 | 4.5e-3 | 1e-3 | 32 | 0.03 | 0.2 | 250K | [0.999, 0.9999] | $[0, \pi/10]$ |
| Sp. Cmds. | 96 | 128 | 8e-3 | 1e-3 | 16 | 0.04 | 0.1 | 212K | [0.1, 0.9999] | $[0, \pi/10]$ |

### B.3 PARAMETERISING AND INITIALISING VARIABLES

Inspired by Orvieto et al. (2023), to ensure $\gamma$ remains in $(0, 1)$ throughout training, instead of directly learning $\gamma$ we learn parameter $\gamma_{\log}$ s.t. $\gamma = e^{-e^{\gamma_{\log}}} \in (0, 1)$. We initialise $\gamma$ to be within the range $[\gamma_{\min}, \gamma_{\max}]$, and $\theta$ within $[0, \theta_{\max}]$. To initialise the input and output matrices $B$ and $C$, we use the Glorot initialisation procedure (Glorot & Bengio, 2010). The weight matrix for the orthogonal $P$ is initialised as a random normal matrix, and we initialise $D$ as a random normal vector applied element-wise to $u_t$.

## C JAX RotRNN IMPLEMENTATION

Here we present a simplified version of the RotRNN layer written in JAX.

```python
import jax
import numpy as np
from jax import numpy as jnp

parallel_scan = jax.lax.associative_scan

def forward(rotrnn_params, input_sequence):
    """Forward pass through the RotRNN layer"""

    thetas, gamma_log, M, B, C, D = rotrnn_params
    gammas = jnp.exp(-jnp.exp(gamma_log))

    T, dim_u = input_sequence.shape

    # compute \xi and normalise B
    B_T_B = jax.vmap(lambda a, b: a @ b)(B.transpose(0, 2, 1), B)
    B_T_B_trace = jnp.trace(B_T_B, axis1=1, axis2=2)
    xi = jnp.sqrt((1 - gammas.squeeze() ** 2) / B_T_B_trace)
    B_norm = jnp.einsum("H, HTD -> HTD", xi, B)

    # create orthogonal matrix P from weight matrix M
    P = jax.scipy.linalg.expm(M - M.transpose(0, 2, 1))

    # project inputs onto heads
    x = jnp.einsum("HDi,Ti->HTD", B_norm, input_sequence)

    # project with P^T
    x = jnp.einsum("HDi, HTi -> HTD", P.transpose(0, 2, 1), x)

    # compute recurrence parallelised over heads
    gammas = jnp.repeat(gammas[:, None], repeats=T, axis=1)
    thetas = jnp.repeat(thetas[:, None], repeats=T, axis=1)
    rec_fn = jax.vmap(
        lambda a, b, c: parallel_scan(binf, (a, b, c)),
        in_axes=(0, 0, 0),
        out_axes=0,
    )
    x = rec_fn(gammas, thetas, x)[2]

    # project back with P
    x = jnp.einsum("HDi, HTi -> HTD", P, x)

    # concatenate heads
    x = x.transpose(1, 0, 2).reshape(T, -1)

    # apply output projection/head mixing and skip connection
    y = jax.vmap(lambda a: C @ a)(x) + D * input_sequence
    return y

def init_params(H, dim_x, dim_u, gamma_min, gamma_max, theta_max):
    """Initialise the learnable parameters"""

    dim_h = dim_x // H

    # random initialisation of \theta in [0, theta_max]
    theta = np.random.uniform(0, theta_max, (H, dim_h // 2))

    # constrained initialisation of \gamma in [gamma_min, gamma_max]
    u1 = np.random.uniform(size=(H, 1))
    gamma_log = jnp.log(
        -0.5 * jnp.log(u1 * (gamma_max**2 - gamma_min**2) + gamma_min**2)
    )
```

```python
    # Glorot initialised input/output matrices
    B = np.random.normal(size=(H, dim_h, dim_u)) / np.sqrt(dim_u)
    C = np.random.normal(size=(dim_u, dim_x)) / np.sqrt(dim_x)

    # Orthogonal weight matrix M
    M = np.random.normal(size=(H, dim_h, dim_h))

    # D is random vector applied element-wise to u
    D = np.random.normal(size=(dim_u))

    return theta, gamma_log, M, B, C, D

def binf(a, b):
    """Binary function for the parallel scan"""
    gamma_i, thetas_i, acc_i = a
    gamma_j, thetas_j, acc_j = b

    # get off diagonal terms [-x2, x1, -x4, x3,...]
    # these will be multiplied by sin(\theta)
    off_diags = jnp.stack([-acc_i[..., 1::2], acc_i[..., 0::2]], axis=-1)
    off_diags = off_diags.reshape(acc_i.shape)

    # duplicate \theta [\theta_1, \theta_1, \theta_2, \theta_2,...]
    theta = jnp.repeat(thetas_j, repeats=2, axis=-1)

    # compute sine and cosine elements of the output
    sin = jnp.sin(theta) * off_diags
    cos = jnp.cos(theta) * acc_i
    acc = gamma_j * (cos + sin)

    return (gamma_i * gamma_j, thetas_i + thetas_j, acc + acc_j)
```

