# OpenReview forum: "RotRNN: Modelling Long Sequences with Rotations"
_ICLR.cc/2025/Conference — ICLR 2025 Conference Withdrawn Submission_

### Official Review · Reviewer_sHHK · 2024-11-02

**Soundness:** 3
**Presentation:** 4
**Contribution:** 1
**Rating:** 5
**Confidence:** 5

**Summary:**

This paper proposes a new class of linear recurrent unit (LRU) which is uprising efficient state-of-the-art model class for long-range sequence modeling. The proposed model, RotRNN, utilizes rotation matrix-based parameterization for state transformation and the explicit normalization method. Based on strong mathematical backgrounds, RotRNN could be simply implemented and strictly guarantees preservation of hidden state magnitude across time-steps. This paper compares RotRNN with former LRU and state space models (SSMs) in theory and experiments to give in-depth understanding for readers.

**Strengths:**

Rigorous mathematical background: The math backgrounds behind the rotation matrix-based parameterization and explicit normalization method are proved with easy-to-read derivations. In addition, those backgrounds lead the simple implementation of RotRNN.

In-depth comparison between former architectures: The theoretic comparisons between RotRNN and (LRU/SSM) are helpful to posit RotRNN within this field.

Strong, latest baselines: This paper compares RotRNN with latest and state-of-the-art baselines (such as S5 and Liquid-S4) in long-range sequence modeling benchmarks.

**Weaknesses:**

Majors:
- Limitation of rotation matrix parameterization: I think there would be drawback with constraining state transformation matrix to be rotation matrix, which might limit expression power of the model.
- Potential drawback of explicit normalization method: It is unclear that whether the explicit normalization method is beneficial for performance. I understand that this method constrains the operation to target a specific range of dependency based on the trained value of $gamma$, so it looks constraining the model’s expressivity. Although it successfully guarantees the converged norm of hidden states during training, its performance-wise effect is not demonstrated in results. An ablation study would be helpful to see its benefit.
- Overall, it is questionable how RotRNN could be advantageous in practice. Comparison of computational efficiency (in terms of FLOPS) would be helpful to show that RotRNN is ‘efficient to compute’.

Minors:
- Weak motivations for rotation matrix parameterization: The three motivations written at the beginning of section 3 aim to simple implementation thanks to the rotation matrix parameterization. However, there is no motivation related to hidden state processing despite of rotation matrix’s unique characteristic (such as regularity). And, it is not clear why real-valued matrix can make it robust to initialization?
- Arguable interpretations of experiment results: The sentence “we find that our model performs best on domains with more discrete input data, such as ListOps, Text and Retrieval, achieving the highest score of all the baselines” seems not clear from the result table. Except ‘Text’ task, Liquid-S4 and S5 achieved the highest scores in ListOps and Retrieval tasks, respectively.
- Confronting claims: This paper argues the unnecessity of SSM model’s theory-based initialization method (HiPPO) in practice (Section 2.3). However, this paper aims to build RotRNN on rigorous theoretic backgrounds. I think those claims are confronting with each other.

**Questions:**

What does $A^{t-k}$ mean in Equation 2?

---

> ### Author Response · Authors · 2024-11-25
>
> Thank you for your feedback on our submission. Please see our specific response to your concerns below, and please see the global comment for general concerns shared amongst reviewers.
>
> - “Limitation of rotation matrix parameterization: I think there would be drawback with constraining state transformation matrix to be rotation matrix, which might limit expression power of the model.”
>     - The reviewer raises a valid question. Our experimental evidence highlights that by constraining the hidden states we achieve competitive performance with prior non-constrained long sequence models. This suggests that constraining the state transition matrix does not limit the expressivity of our model. Moreover, there is theory surrounding free-groups of rotation matrices [1] that suggests that rotations could be general enough to uniquely encode arbitrary sequences of inputs. However, to unlock to full encoding power of these methods, one may need to turn to input-dependent transformations, as is seen in Mamba and Griffin, for example. However, as this is a significant extension to the vanilla RotRNN algorithm, we leave this study for future work.
> - “Potential drawback of explicit normalization method: It is unclear that whether the explicit normalization method is beneficial for performance. I understand that this method constrains the operation to target a specific range of dependency based on the trained value of gamma, so it looks constraining the model’s expressivity. Although it successfully guarantees the converged norm of hidden states during training, its performance-wise effect is not demonstrated in results. An ablation study would be helpful to see its benefit.”
>     - In practise we find that our normalisation scheme successfully prevents blow-ups in the magnitude of the hidden state as well as preventing vanishing gradients, whilst also achieving competitive performance on long sequence modelling tasks. We believe this to be indicative of our models expressivity. We do however appreciate the reviewers concerns about a lack of experimental evidence and will conduct further ablations in future revisions to further evaluate the impact of normalisation on our models expressivity.
> - “Confronting claims: This paper argues the unnecessity of SSM model’s theory-based initialization method (HiPPO) in practice (Section 2.3). However, this paper aims to build RotRNN on rigorous theoretic backgrounds. I think those claims are confronting with each other.”
>     - We thank the reviewer for raising this point of concern. We don’t argue that correct initialisation is not necessary, but instead that the HiPPO initialisation scheme is overly complicated and that simpler schemes, which can still be rigorous, can be used instead. In future revisions of our work we will ensure to make this point clearer.
>
> [1] De Groot, J. and Dekker, T. (1954). Free subgroups of the orthogonal group. Compositio Mathematica, 12:134–136.

---

### Official Review · Reviewer_PLUo · 2024-11-03

**Soundness:** 3
**Presentation:** 3
**Contribution:** 2
**Rating:** 3
**Confidence:** 3

**Summary:**

State-based models have reduced the long-path problem of traditional recurrent models, but require complicated initialization procedures in order to avoid vanishing/exploding gradients.  This paper demonstrates a new linear recurrent unit in which the recurrence is constrained to be a rotation, scaled by a head-wise decay constant.  The head-wise decay constant is matched to the amplitude of the input coefficients, in order to guarantee that the magnitude of the state vector remains constant over time, avoiding the vanishing/exploding gradient problem without complicated initialization procedures.

**Strengths:**

The factorization of the linear recurrence matrix into cosine and sine rotations is elegant, and was a pleasure to read.

**Weaknesses:**

The key weakness of this paper is a minor oversight in the analysis of LRU, which calls into question the value of this paper's contribution.

The proposed algorithm is very similar to LRU, except that it forces the eigenvalues of the recurrence matrix to come in complex conjugate pairs.  The manuscript notes this as a weakness of LRU: that LRU does not require the eigenvalues to come in complex conjugate pairs, and instead, LRU simply takes the real part of the output of the linear layer.   It seems that the authors of this paper do not realize that taking the real part of the output of the network solves the problem of arranging eigenvalues into complex conjugate pairs, because the real part of any complex number is the average of the complex number and its complex conjugate: Re(z) = (z+z*)/2.  Thus, if z(t)=P Lambda^t P^T z(0), and if z(0) is real, then Re(z) = (1/2)(P Lambda^t P^T + P* Lambda*^t P*^T) z(0).  By taking the real part of the output, LRU is pairing each explicit eigenvalue with an implicit eigenvalue equal to the complex conjugate of the explicit eigenvalue, thus effectively doubling the dimension of the recurrent layer, at no extra computational cost.

Since LRU already has each eigenvalue paired with an implicit conjugate eigenvalue, the only remaining difference I can see between the proposed algorithm and LRU is the proposed input normalization, which guarantees that the recurrent state vector maintains constant norm.  In theory, this seems like a useful contribution, since it explicitly avoids gradient vanishing/explosion.  In practice, it's not clear that LRU suffers from problems of gradient vanishing or explosion.  In the example shown, the norm of LRU gets large, but it never seems to overflow.  The proposed algorithm has better performance than LRU on a toy example, but it is not clear that the difference is statistically significant.

**Questions:**

Given that the real-part operation at the output of LRU implicitly pairs each eigenvalue with its complex conjugate, what are the remaining important differences between the proposed algorithm and LRU?

Which of the performance differences shown are statistically significant?

---

> ### Author Response · Authors · 2024-11-25
>
> Thank you for your feedback on our submission. Please see our specific response to your concerns below, and please see the global comment for general concerns shared amongst reviewers.
>
> - “By taking the real part of the output, LRU is pairing each explicit eigenvalue with an implicit eigenvalue equal to the complex conjugate of the explicit eigenvalue, thus effectively doubling the dimension of the recurrent layer, at no extra computational cost.”
>     - We thank the reviewer for clarifying this insight into the workings of LRU. Our broader point about the drawbacks of the LRU was not the way that it works in practice, but that it does not directly follow from its theoretical derivation. In particular, we highlight two ways which the practical implementation of the LRU deviates from what would be necessary if the hidden state matrix was indeed derived from the eigendecomposition of a real-valued matrix A. Firstly, the eigenvalues of Lambda would need to come in complex conjugate pairs, which we agree can be solved in practice by implicitly averaging the output and its complex conjugate. However, the more significant point regarding the deviation from theory caused by taking the real part of the output is that the complex parts of the \tilde{C} and \tilde{B} matrices are not enforced to be consistent. Therefore, it is likely that the P and P^-1 components of these matrices are not in fact the inverses of each other, which would be required if the hidden state matrix was in fact the eigendecomposition of a real-valued matrix A, as posited in the LRU paper. Taking the real value of the output may be a practically useful way of circumventing this problem, but we do not find it entirely satisfactory from a theoretical standpoint. We did not mean to conflate these two points in the paper, and future revisions will endeavour to make this more clear. Our aim for RotRNN was to provide a simple algorithm with a straight forward but rigorous theoretical background, whose implementation remains faithful to its derivation throughout the entirety of training.

---

### Official Review · Reviewer_oKdr · 2024-11-05

**Soundness:** 2
**Presentation:** 3
**Contribution:** 2
**Rating:** 3
**Confidence:** 2

**Summary:**

The paper presents a linear state space model, where the recurrent state is transformed by a rotation matrix, A, that is an exact rotation matrix, and input is transformed by an input matrix. In prior works like LRU, careful initialization is performed to ensure that the rotation matrices at the start of learning are orthogonal and that outputs of the state space model are real values . However, as training proceeds, it is possible that complex parts arrive and are ignored. In this paper, the authors propose RotRNN - a parameterization of rotation matrices that is exact (although it doesn't cover the space of all rotation matrices, if I understood correctly). To do this, the authors show that any general matrix M can be smoothly mapped to the special orthogonal group by taking the matrix exponential of M-M^t.  Now, P = exp(M-M^T) can be mapped to the special orthogonal group rotation matrix A, through a block diagonal matrix $\theta$ by A= $ P \theta P^t$.
Using this scheme orthogonal recurrent rotation matrices can be generated.

Aside from ensuring that an orthogonal rotation matrix can be produced, the authors also ensure that the hidden state $x_t$ is well behaved an preserves a constant norm in expectation. This is performed by using a normalization constant $\alpha$ to counteract the decay factor $\gamma$ applied to the rotation of the recurrent states. In practice this is actually done by rescaling the output of the application of the input matrix to the input.


Results of using the method are shown on LRA benchmarks and on Raw Speech classification using the Speech Commands classification task.

**Strengths:**

While I didn't look at the detailed linear algebra proofs, they seemed correct to me mathematically and made sound intuitive sense. There are also results showing that norms of the state are well preserved when the model is run, so that part of the proposal also seems to work well. The provided Jax code also makes it clear to see how the implementation matches the technical details of the paper.

**Weaknesses:**

While I am moved by the simplicity of their parameterization compared to prior works, I am not sure if the contribution is enough to merit a paper in ICRL with the kind of experimental exploration performed. I think a proper paper would run much further with the proposed method than the author(s) have done here. Speech commands is quite a small dataset and the results on it, and on LRA shed little light into the details of their method. And the results on these datasets are not necessarily better than prior methods. So the selling point of the method from a standpoint of improved results over LRU is not clear. Furthermore, other than showing the norms of the states are well behaved, the paper does not offer more technical insight either, for others to build upon. Without that, this is more of an exposition of a mathematical trick rather than a full contribution.

**Questions:**

How well do you think this model can scale up, compared to other approaches ?
Does it interact well as a preprocessing layer into transformer based models ?
The block diagonal matrices look like blocks of rotations in 2D. Are these the only kinds of block rotation matrices that are possible for theta ?
Is it possible to control the periodicity of rotations using the block diagonal 'blocks' in some way ?

---

> ### Author Response · Authors · 2024-11-25
>
> Thank you for your feedback on our submission. For our response to your concerns, please see the global comment.

---

### Official Review · Reviewer_bBq2 · 2024-11-07

**Soundness:** 3
**Presentation:** 2
**Contribution:** 2
**Rating:** 5
**Confidence:** 3

**Summary:**

This paper proposes a new linear recurrent model using rotation matrices. The aim of introducing rotation matrices is to enforce the theoretical constraints of LRU that are missing in its practical implementation. Parametrizing recurrent state matrix as rotations allows the authors to present a fast method of matrix powers and also allows then to present a normalization process that helps near constant hidden state norms, both of these are essential for linear RNNs especially in long sequence domain.

**Strengths:**

*  Good theoretical justification of parametrizing recurrent state matrix as rotation matrix:
    * Show how rotations can be easily decomposed for efficient matrix power computation
    * Show how orthogonality of rotation matrices is used for normalization, leading to almost constant hidden state norms

* Very effective normalization enabled by the orthogonality of rotation matrices (as seen in figure 3) ensuring that hidden state norms do not vanish/explode across long sequences, which is important for stable training.

* Reproducibility: detailed hyperparameters and jax code are provided for reproducibility.

**Weaknesses:**

* The proposed approach does not seem to improve the performance on the LRA benchmarks. As shown in the table 1, the proposed approach is better than baselines only on text and that too with a very small margin. While on other benchmarks and on average, it performs significantly worse (upto 10 percent points in case of Path-X).

* Also on speech commands classification task of table 2, the proposed approach is not better than any of the shown baselines.

* Although the proposed implementations using rotation matrices in this paper enables theoretical constraints of LRU, they don’t improve the performance on downstream tasks. Also, the paper does not show how efficient the proposed method is compared to the baselines in terms of compute/resource use.

**Questions:**

The authors never concretely show (in terms of any sensible metrics, such as GPU hours etc) how efficient their model is compared to the baselines. It would be great if they could do a more thorough job at comparing their models against the baselines. The value of this question is particularly important, especially since their model performance is not significantly better than the baselines. Why wasn't their model compared rigorously in terms of efficiency?

---

> ### Author Response · Authors · 2024-11-25
>
> Thank you for your feedback on our submission. For our response to your concerns, please see the global comment.

---

### Author Response · Authors · 2024-11-25

We appreciate all reviewers concerns about limited experimental investigation into the efficacy of our model and its specific design components, such as normalisation scheme and use of rotation matrices. Whilst our current evaluation considers standard long range sequence modelling tasks, for which RotRNN is competitive with prior works, we appreciate the desire for a greater range of experimental results. However, please note that the goal of this paper was to show that performance of prior works which use complicated initialisation procedures can be matched with a conceptually simpler model, rather than to achieve state of the art results. In addition, we aimed to present strong theoretical motivation for our design choices and an implementation that remains faithful to its theoretical derivation. In future revisions we will ensure to include a wider range of experimental evidence highlighting the benefits of our parameterisation, including performance on different benchmarks, efficiency comparisons with other models and expressivity comparisons with unconstrained linear RNNs.

---

### Note · Authors · 2024-11-25

**Comment:**

We highly appreciate the feedback on our submission from our reviewers. Given the reviewers' concerns regarding lack of experimental evidence to investigate the performance and design choices of our algorithm, we have decided to withdraw our submission and will aim to improve on these concerns in future revisions.

**Withdrawal Confirmation:**

I have read and agree with the venue's withdrawal policy on behalf of myself and my co-authors.